# The Mediating Role of Perceived Comfort between Workplace Attachment Style and Perceived Stress

**DOI:** 10.3390/ijerph20075377

**Published:** 2023-04-03

**Authors:** Justine Rebillon, Olivier Codou, Jean-Félix Hamel, Eva Moffat, Fabrizio Scrima

**Affiliations:** 1Department of Psychology, University of Rouen Normandy, 76130 Mont-Saint-Aignan, France; 2Department of Social and Economic Administration, Center for Studies and Research on Organizations and Strategy, University of Paris Nanterre, 92001 Nanterre, France

**Keywords:** attachment theory, place attachment, workplace attachment, comfort, stress

## Abstract

Past studies highlight the relevance of attachment theory to the study of workplace stress and the impact of employee assessments about the physical–spatial work environment on their health. This paper is one of a number of works studying the points of connection between Bowlby’s attachment theory and the place attachment theory adopted by environmental psychologists. We proposed that a secure workplace attachment style would be negatively associated with perceived stress (and vice versa for insecure workplace attachment styles). Perceived comfort was hypothesized to mediate these effects. A convenience sample of French white-collar workers (*N* = 379) completed an online survey. Hypotheses were tested using the PROCESS macro. Both insecure workplace attachment styles (i.e., avoidant and preoccupied) were negatively associated with perceived comfort, which partially mediated their positive effect on perceived stress. The preventive influence of a secure workplace attachment on perceived stress was entirely mediated by its positive effect on perceived comfort. By setting different expectations regarding the work environment, workplace attachment styles could translate into a more or less stressful and comfortable employee experience. The more secure the bond employees internalize with their workplace, the more they might benefit from its comforts’ restorative potential.

## 1. Introduction

Work can affect employees’ physical and mental health in a number of ways (e.g., work-related injuries or burnout). Typically, poor working conditions and/or poor work organization can lead to a variety of negative consequences [1,2,3]. For example, several studies have shown that poor working conditions can lead to musculoskeletal disorders [4] as well as depression [5]. Past work also focused on uncovering how the work environment and its physical-spatial characteristics can impact stress at work [6,7]. For instance, office temperature [8], office type [9], or workplace design [10] can positively or negatively impact employee’s stress. Perceived stress is defined as the degree to which situations in a person’s life are evaluated as overwhelming, unpredictable, and uncontrollable [11]. Scarce until the 1980s, studies on working conditions (and specifically that of office environments) have been multiplying in the past twenty years [12]. They have generally shown that the majority of employees continuously assess their work environment and are aware of most of its impacts on their health. Moreover, employees who self-assess that they work in a healthy work environment have higher job satisfaction, morale and commitment, and lower absenteeism and intention to leave the organization [13]. However, it is still unclear whether or not this assessment process may impact their perceived stress (and if so, how).

In an attempt to further understand how the assessment of certain physical–spatial characteristics of work environments may impact various work outcomes, recent work in environmental psychology has notably relied on the construct of place attachment [14]—i.e., an affective bond that develops from the interaction between individuals and places that are meaningful to them; as such, the workplace has been studied as an object of attachment [15]. Taking advantage of the conceptual reciprocities between place attachment theory and Bowlby’s attachment theory [16], a novel stream of research has developed around their integration, coining constructs such as place attachment styles [17] or—more specific to our purposes—workplace attachment styles [18]. In light of these elements, this paper has two main objectives. As it has not yet been examined in the literature, we first aim to investigate the relationship between workplace attachment styles and perceived stress. Second, we intend to assess the mediating role of employees’ assessment of the quality of their work environment between these variables—which we operationalized as their satisfaction with perceived comfort. From a theoretical standpoint, this study will thus contribute to the large body of work in environmental psychology acknowledging the workplace and its design as resources which may help employees cope with stress.

### 1.1. Workplace Attachment Styles and Perceived Stress

Defined as an affective bond between an individual and a specific place [14], the first studies on place attachment date back some 60 years [19]. Over the years, many comparisons have been made between the theory of place attachment and classical attachment theory [16,20]. Points of contact between the two include several physical, social and emotional components [21]. Scannell and Gifford [17] notably extend this comparison by placing need satisfaction (i.e., the primary role of the attachment figure) at the center of this reflection. Just as children seek closeness and safety in the event of separation from the attachment figure [22], adults activate a series of behavioral patterns to maintain proximity with a place that satisfies their needs [23].

Indeed, past studies suggest many places can be considered as objects of attachment: the home [24], the neighborhood [25], the city [26], national parks [27] and also the workplace [15]. Drawing from Bartholomew and Horowitz’s model [28], Scrima et al. [18] proposed the existence of possible place attachment styles stemming from positive or negative evaluations of both the Self and the workplace. In this context, a positive Self is an internalized view of self-esteem that is not dependent on the evaluations of others, whereas a negative Self is an internalized view of anxiety about acceptance and fear of rejection [29]. Individuals tend to develop a positive or negative vision of a place depending on its relative ability to adequately meet their needs [30].

While a secure workplace attachment style is the result of a positive representation of both self and the workplace, the presence of any negative representation suggests an insecure form of attachment. A positive representation of the Self and a negative representation the workplace characterizes an avoidant workplace attachment style, whereas the inverse configuration (i.e., negative self and positive workplace) designates a preoccupied (or anxious) workplace attachment style. Finally, carrying negative representations of both the Self and the workplace is associated with a disorganized workplace attachment style [18]. However, disorganized attachment is often associated with severe psychopathological disorders [31,32,33] and therefore seldom found among samples of employed people.

To date, no articles have associated workplace attachment styles and perceived stress in employees. There is empirical evidence, however, that a place can act as a safe haven [34], and/or play a restorative role [35,36] that allows for stress relief [37]. Given the positive nature of the underlying affective bond, it could be hypothesized that employees with a secure workplace attachment style are able to take advantage of the benefits offered by the workplace to buffer the adverse effects of perceived stress.

According to Berto [38], the physical environment can impair or enhance stress-coping strategies. In line with this proposition, a variety of research has shown the restorative role of the workplace [39] as well as how such an environment may reduce perceived stress in employees [40]. Employees with an avoidant attachment style have a negative view of the workplace—that is, they do not recognize it as capable of meeting their needs [18]. By adopting congruent behaviors—such as physically avoiding it—avoidant employees may be unable to use the workplace as a resource to cope with perceived stress.

Within the paradigm of Bowlby’s attachment theory [16], Richards and Schat [41] found that preoccupied employees have a strong tendency to seek instrumental and emotional support from their attachment figure; this may be explained by the anxiety underlying the relationship due to their negative self-representations. Paraphrasing this idea, although employees with a preoccupied workplace attachment style maintain a positive view of the workplace (i.e., the object of attachment), they see themselves as unable to take advantage of the workplace to meet their needs or as unworthy of such care [18]. In other words, despite their attempts to seek restorative support in the workplace, preoccupied employees may believe they do not possess the tools or do not deserve to benefit from it, which may foster a state of distress. Consistent with the cited literature, it is hypothesized that:

**H_1a_:** *Secure workplace attachment is negatively related to perceived stress*.

**H_1b_:** *Avoidant workplace attachment is positively related to perceived stress*.

**H_1c_:** *Preoccupied workplace attachment is positively related to perceived stress*.

### 1.2. The Mediating Role of Perceived Comfort

According to Vischer [42], workplace comfort is organized in three hierarchical dimensions: physical, functional and psychological. Physical comfort refers to compliance with hygiene and safety standards. Functional comfort focuses on the evaluation of the building’s performance. Finally, psychological comfort includes psychosocial aspects such as a sense of belonging or control over the work environment [43]. Work environments are inherently instrumental spaces that must satisfy specific functionalities through physical and psychosocial characteristics [44].

In the present work, we focused on satisfaction about employees’ perceived comfort. There is empirical evidence indicating that the intensity of attachment to a place can influence an individual’s perception of it. For example, the greater the place attachment, the less it is perceived as polluted [45] or the lower the perceived risk of unlikely negative events occurring [46]. Using a semi-longitudinal model, Scrima et al. [47] observed that workplace attachment can impact levels of job satisfaction but not vice versa. However, very little attention has been dedicated to the relationship between workplace attachment styles and perceived comfort. Scrima et al. [48] showed that a secure workplace attachment style is positively associated with design satisfaction. In a recent paper, secure workplace attachment style was shown to be positively associated with the perception of physical–spatial comfort [49]. To our knowledge, however, no research has investigated the relationship between insecure workplace attachment styles and perceived comfort. As a way to maintain a sense of consistency between the underlying view and the physical characteristics of the workplace [50], an avoidant workplace attachment style may negatively influence perceived comfort. Indeed, adult attachment studies in the marital [51], sexual [52] or parental contexts [53] suggest avoidant individuals tend to be generally dissatisfied with their attachment figure.

In spite of their favorable view of the workplace, employees with a preoccupied workplace attachment style still fail to relate positively to their object of attachment overall due to their negative self-representations and the strain it may foster [50]. The insecure nature of the relationship—that is, their constant need to seek proximity [54] and support [55]—may also translate to their environment being experienced as less comfortable. Regarding workplace attachment styles and perceived comfort, we hypothesize that:

**H_2a_:** *Secure workplace attachment is positively related to perceived comfort*.

**H_2b_:** *Avoidant workplace attachment is negatively related to perceived comfort*.

**H_2c_:** *Preoccupied workplace attachment is negatively related to perceived comfort*.

Surprisingly little research has explored if and how perceived comfort may relate to employee stress [56]. In 2008, Rashid and Zimring published a literature review showing the various impacts the physical–spatial characteristics of the office may have on employees’ perceptions, affects and behaviors [57]. As individuals usually spend 20 to 60 h per week in the workplace, it is no surprise its social and physical factors can affect their well-being or stress levels [58]. For example, Sundstrom et al. [59] identified noise as one of the major environmental stressors for employees working in open-plan environments. On the other hand, according to the Attention Restoration Theory, green spaces in the office can reduce stress levels [60]. Some research has also shown that—in addition to the physical–spatial characteristics of the place—the satisfaction toward these elements may impact the level of well-being of individuals. For example, Gilchrist, Brown and Montarzino [61] found that having a window with a view of nature does not impact employee well-being, whereas satisfaction toward the view statistically improves it. Finally, Scrima et al. [48] found that satisfaction with design significantly reduces employee exhaustion. Thus, it is possible to hypothesize that:

**H_3_:** *Perceived comfort is negatively related to perceived stress*.

**H_4a_:** *Perceived comfort mediates the relationship between a secure workplace attachment and perceived stress*.

**H_4b_:** *Perceived comfort mediates the relationship between an avoidant workplace attachment and perceived stress*.

**H_4c_:** *Perceived comfort mediates the relationship between a preoccupied workplace attachment and perceived stress*.

## 2. Materials and Methods

Data were collected in a convenience sample of 379 white-collar workers. They were asked to fill out an anonymous online questionnaire regarding the investigation of the relationship between some psychosocial variables and workers’ well-being. Employees were recruited online through posts on various professional (e.g., LinkedIn) and nonprofessional (e.g., Facebook) social media platforms. To access the questionnaire, they had to confirm they had been working in the same organization for at least a year. The sample was predominantly female (67.8%) and aged 19–64 years (*M* = 42.19, *SD* = 11.09) with a length of service in the current organization ranging from 1 to 40 years (*M* = 10.99, *SD* = 9.04). Regarding organizational status, 75.5% were clerks, 14.5% were executives and the remaining 10% were managers. After giving their informed consent, participants had to report working in the same organization for at least one year. A post hoc power analysis was conducted using the following parameters: 6 predictors, 379 participants, medium effect size (*f*² = 0.15) and α = 0.05. We achieved a satisfactory level of power (*β* = 0.99).

The Workplace Attachment Style Questionnaire (WASQ [62]) was used to measure employees’ workplace attachment style. This scale identifies three styles: avoidant (e.g., “In my organization, I prefer to avoid certain places, even if that interferes with my work”), secure (e.g., “I enjoy the time that I spend in my workplace”) and preoccupied (e.g., “Just thinking about my workplace makes me feel anxious”). Each factor comprises 5 items with a 7-point Likert response scale from “Totally disagree” to “Totally agree”. Before responding to the items, employees were given the instruction to first think about their workplace, its rooms and corridors, the color of its walls, its sounds, noises, and smells and the people with whom they usually share it [63]. In the present study, the original correlated three-factor structure showed satisfactory fit indices (*χ*²/*df* = 2.97, *CFI* = 0.96, *NNFI* = 0.95, *SRMR* = 0.05). Measured using McDonald’s omega, the internal consistency of avoidant, secure and preoccupied attachment style subscales was satisfactory (0.83, 0.87 and 0.88, respectively).

Perceived comfort was measured using the Satisfaction with the Work Environment Scale [44]. This instrument consists of 14 items divided into 2 subscales measuring environmental comfort (e.g., air circulation) and control/privacy (e.g., isolating oneself from the gaze of others). Participants were asked to respond according to how satisfied they are using a 5-point Likert scale from 1 “Not at all satisfied” to 5 “Completely satisfied”. In line with the results of Fleury-Bahi and Marcouyeux [44], a model with two latent factors plus a second-order factor was tested and provided satisfactory fit indices (*χ²*/*df* = 2.93, *CFI* = 0.95, *NNFI* = 0.92, *SRMR* = 0.05). The overall perceived comfort factor obtained an adequate internal consistency index (*ω* = 0.88).

Perceived stress was measured using the French version [64] of the Perceived Stress Scale [11]. This scale consists of 10 items (e.g., “In the last month, how often have you felt that you were unable to control the important things in your life?”) that saturate on a single latent factor. Participants responded via a 5-point Likert scale from 1 “Never” to 5 “Very often”. Confirmatory factor analysis supported the expected single-factor model (*χ²*/*df* = 2.93, *CFI* = 0.96, *NNFI* = 0.93, *SRMR* = 0.05) (*ω* = 0.76).

## 3. Results

### 3.1. Preliminary Analysis

According to Podsakoff et al. [65], cross-sectional designs could be influenced by the common method bias. In an attempt to control for such bias, we compared six models using the AMOS 4.0 software (IBM, New York, NY, USA.) [66]. In Model 1, all items of the five variables saturated on a single latent factor. Model 2 had two covaried latent factors: the first included all the items of the Workplace Attachment Style Questionnaire, the second included the items of the scale on perceived comfort and stress. Model 3 had three covaried latent factors—i.e., one for each construct. In the fourth model (Model 4), items measuring workplace attachment styles were divided among two factors representing secure and insecure forms of attachment. In Model 5, each workplace attachment style (i.e., secure, avoidant and preoccupied) had its dedicated latent factor, for a total of five covaried latent factors. In the last model (Model 6), a method latent factor on which all items saturated was added to the measurement model. The results (Table 1) indicate that the only model showing satisfactory fit indices is model 5. As they are not nested, models 5 and 6 cannot be compared using *χ*²; however, the fact that other fit indices such as CFI did not improve (Δ < 0.01) indicates that common method bias did not substantially influence our data [67].

### 3.2. Descriptive Statistics

Table 2 shows the descriptive statistics, correlations and omega indices of the variables in this study. Skewness (S) and kurtosis (K) indices are between –1.11 and 1.27, indicating a relative absence of normality violation. The avoidant attachment style to the workplace appears to be negatively correlated with perceived comfort (*r* = −0.51, *p* < 0. 001) and positively correlated with perceived stress (*r* = 0.37, *p* < 0.001). A secure workplace attachment style is positively correlated with perceived comfort (*r* = 0.56, *p* < 0.001) and negatively with perceived stress (*r* = −0.19, *p* < 0.001). A preoccupied workplace attachment style seems to show the same pattern as its avoidant counterpart—i.e., negatively correlated with perceived comfort (*r* = −0.41, *p* < 0.001) and positively with perceived stress (*r* = 0.38, *p* < 0.001). Finally, perceived comfort is negatively correlated with perceived stress (*r* = −0.27, *p* < 0.001).

### 3.3. Hypothesis Testing

To test our hypotheses, three mediation analyses were performed (one for each attachment style) using the PROCESS macro [68] and in particular using model 4; 5000 bootstrap samples were used to determine 95% confidence intervals of effects. Before performing the analyses, all variables were standardized. As shown in Table 3, secure workplace attachment is positively associated with perceived comfort (*β* = 0.55, LL = 0.47, UL = 0.64), but not with perceived stress (*β* = −0.04, LL = −0.15, UL = 0.17). Perceived comfort is negatively associated with perceived stress (*β* = −0.22, LL = −0.34, UL = −0.11). Furthermore, the bootstrapped 95% confidence interval of the indirect effect (*β* = −0.13, LL = −0.19, UL = −0.06) indicates a total mediation effect of perceived comfort on the relationship between secure workplace attachment and perceived stress.

In the model with avoidant workplace attachment as an independent variable (Table 4) we observed this type of bond to be negatively associated with perceived comfort (*β* = −0.50, LL = −0.58, UL = −0.41) and positively associated with perceived stress (*β* = 0.28, LL = 0.17, UL = 0.38). Perceived comfort is negatively associated with perceived stress (*β* = −0.11, LL = −0.21, UL = −0.01). The bootstrapped 95% confidence interval of the indirect effect indicates a partial mediating effect of perceived comfort on the relationship between an avoidant workplace attachment and perceived stress (*β* = 0.05, LL = 0.00, UL = 0.11).

In our third and final model, the preoccupied workplace attachment was the focal predictor. The results (Table 5) appear similar to those obtained for the avoidant workplace attachment style. A preoccupied workplace attachment style is negatively associated with perceived comfort (*β* = −0.41, LL = −0.50, UL = −0.32) and positively associated with perceived stress (*β* = 0.32, LL = 0.22, UL = 0.41). Perceived comfort is negatively associated with perceived stress (*β* = −0.11, LL = −0.21, UL = −0.02). The indirect effect indicated by the bootstrapped 95% confidence interval is statistically significant (*β* = 0.05, LL = 0.00, UL = 0.09), suggesting a partial mediation effect.

## 4. Discussion

In spite of the flourishing literature integrating place attachment and attachment theory in the context of work [48,63], no studies to date have questioned if and how the quality of the affective bond between employees and their workplace might impact stress. To begin this process, the present study had two objectives. First, we investigated the relationship between workplace attachment styles (i.e., secure, avoidant and preoccupied) and perceived stress; our hypotheses suggested that a secure bond may allow the workplace to play a stress-relieving role, whereas insecure forms of attachment would be associated with more strenuous interactions. We also examined the predicting role of these working relationship models regarding perceived comfort—i.e., the extent to which the workplace satisfies employees’ environmental needs as well as the privacy and control it affords them. Insecure workplace attachment styles are defined by a lack of confidence regarding the workplace’s ability to meet employees’ needs (i.e., avoidant) or one’s ability to take advantage of it to do so (i.e., preoccupied) [18]; thus, we proposed that the more employees would report such bonds, the less comfortable they would feel (and vice versa regarding secure workplace attachment). Second, in line with previous studies, we suggested such comfort may in turn negatively predict perceived stress, thereby mediating the relationship between workplace attachment styles and perceived stress.

Regarding the first set of hypotheses, our results suggest that insecure forms of workplace attachment are associated with more perceived stress (H_1b_, H_1c_); this is consistent with previous research suggesting that—whether by viewing their work environment as hostile and unable to fulfill their needs (i.e., avoidant) or themselves as incapable of making good use of its features (i.e., preoccupied)—employees with such relationship models tend to perceive more stressors in the work environment [50]. In fact, these forms of attachment are often discussed through the scope of the associated undesirable effects such as stress; for instance, Johnstone and Feeney [69] concluded that attachment anxiety led to less adaptive coping strategies, whereas Ronen and Baldwin [70] observed that perceived stress mediated its relationship with burnout.

However, insecure forms of workplace attachment may also be understood as the result of an adaptive mechanism internalized following unresolved threatening experiences in the workplace [71]. Some authors argue that individuals with insecure forms of attachment may provide groups with advantages such as heightened threat-detection in a given environment [72]. So as to clarify the role of workplace attachment styles on employee stress, future work may want to investigate this proposition in the organizational context—by combining individual and team-level measurements, for instance. Nonetheless, our results reaffirm that both attachment theory and place attachment literature are indeed relevant to the study of organizational phenomena, including stress in the workplace. All other things being equal, we did not detect the hypothesized negative direct effect between secure workplace attachment and perceived stress (H_1a_); however, secure workplace attachment can be associated with less perceived stress (Table 2), which supports the notion that internalizing the physical workplace as a safe space might reduce stress perception by another mechanism—such as shaping the assessment employees make of the comfort provided by this environment.

As expected, a secure workplace attachment was positively associated with perceived comfort (H_2a_), whereas insecure types of workplace attachment were negatively associated (H_2b_, H_2c_). The first of these results echoes recent work by Mura et al. [49], which proposed secure workplace attachment as a predictor of perceived comfort—in short, the more employees have internalized their work environment as a place which can meet their needs, the more satisfied they are with the physical comfort it provides. Our work is the first to date to provide empirical evidence that the opposite may be true of employees with insecure workplace attachment styles—that is, the more insecure the relationship, the less comfortable the work environment is perceived to be. These results may all be explained by a number of well-established psychological phenomena including selective exposure—focusing on information in one’s environment that is congruent with one’s attitudes so as to avoid cognitive dissonance—and/or confirmation bias [73]. In summary, employees may tend to filter and interpret information regarding their workplace in a way that is consistent with their internalized working model of the underlying relationship.

Our data supports a negative relationship between perceived comfort in the workplace and perceived stress; as suggested by past research, employee satisfaction regarding the physical elements of their work environment may play a protective role regarding perceived stress [40,74]. Bootstrapping results also indicate the indirect effect of workplace attachment style on perceived stress via perceived comfort is significant in all mediation models, which provides support for our final hypothesis (H_3_). In the case of secure workplace attachment, we observed the total mediation (H_4a_) of a negative effect, suggesting the increase in perceived comfort as the underlying stress-reduction mechanism. Conversely, the indirect effect of insecure forms of workplace attachment on perceived stress (H_4b_, H_4c_) was only partially mediated and positive. Due to the counter-dependent or anxious nature of the underlying relationship, an insecure workplace attachment makes for a more stressful experience overall [48]; however, this process may be partially encouraged by the general dissatisfaction it seemingly fosters for the physical elements of the work environment. These findings are in line with past work emphasizing the protective and restorative role of secure place attachment [75].

The present study has several limitations. First, we collected data from a convenience sample of employees from multiple organizations, which did not allow for the control of certain environmental variables; for instance, different office designs may influence employee satisfaction regarding their level of comfort [76]. Additionally, pooling executives and managers with the clerks of our sample may have inadvertently introduced some bias; supervisors tend to have more functional control over the physical workspace, whereas employees (such as the clerks in our sample) grow more dissatisfied as supervisor control increases [77]. Thus, future work may need to confirm whether the reported effects are valid across these different groups. Second, the cross-sectional design of this study is far from optimal regarding the testing of mediation hypotheses [78]. Although the direction of the tested effects was justified based on the relevant scientific literature, we remain unable to firmly establish causality; therefore, future work should turn to longitudinal or experimental designs in order to clarify the directionality of these effects. Finally, although the French version of the Perceived Stress Scale was validated in a sample of workers [64], its scope appears larger than that of psychological distress in the workplace. Future studies may obtain more accurate and generalizable results by using a contextualized measurement tool.

Beyond these limitations, our work has several larger implications for organizations and managers to consider. First, identifying the type of affective bond employees have developed with their workplace (i.e., their workplace attachment style) may help determine who among them could most benefit from stress prevention and related interventions [69]; it is also possible such attention from typical attachment figures such as managers [79] may contribute to these employees internalizing more secure representations of the workplace, thereby creating a virtuous circle regarding stress prevention. Second, while a more comfortable workspace remains desirable, it is important for organizations to consider what lies in the gap between objective features and comfort perceptions [80]; psychological constructs such as workplace attachment style could play a part in shaping employees’ expectations regarding the workplace, which in turn may influence stress perceptions accordingly. When it comes to stress prevention, the relative beneficial or detrimental role of creature comforts seems to depend on the extent to which employees consider the workplace a comfort zone (and the expectations so implied).

## 5. Conclusions

Drawing from attachment theory, place attachment and environmental psychology literature, this study examined the mediating role of perceived comfort between workplace attachment styles (i.e., secure, avoidant and preoccupied) and perceived stress in a sample of French white-collar employees. Building on previous work, these results confirm that environmental comfort in the workplace tends to have a preventive effect on stress perceptions. Additionally, we are the first to show that the more insecure the relationship between employees and their workplace, the more stressed they tend to feel. Moreover, our results indicate that the relieving effect of attachment security on perceived stress is entirely mediated by perceived comfort, whereas the detrimental effect of insecurity is only partially so. We contribute to the literature suggesting that the type of affective bond employees develop with their workplace can predict stress and provide the first empirical evidence regarding the underlying psychological mechanism. Finally, our work may be useful to managers and practitioners seeking to use the work environment as a stress prevention tool, as well as identifying who may most benefit from such interventions. Specifically, we suggest management should provide employees with a place that satisfies both their demands in terms of environmental comfort and is also capable of meeting their various needs; this would allow employees to internalize positive self-representations, which may in turn contribute to the development of a more secure attachment to the workplace.

## Figures and Tables

**Table 1 ijerph-20-05377-t001:** Measurement model.

Model	*χ*²	*df*	*p*	*χ*²/*df*	*CFI*	*NNFI*	*SRMR*	Δ*χ*^2^	Δ*df*	*p*
Model 1	2686	669	<0.001	4.02	0.73	0.70	0.09			
Model 2	2389	668	<0.001	3.58	0.78	0.74	0.08	297	1	<0.001
Model 3	1959	666	<0.001	2.94	0.82	0.81	0.08	430	2	<0.001
Model 4	1487	663	<0.001	2.24	0.89	0.88	0.07	472	3	<0.001
Model 5	1348	659	<0.001	2.04	0.91	0.90	0.06	139	4	<0.001
Model 6	1348	658	<0.001	2.05	0.91	0.90	0.06	-	-	-

**Table 2 ijerph-20-05377-t002:** Means, standard deviations, skewness, kurtosis and zero-order correlations (ω on the diagonal).

	Mean	95% CI (LL UL)	SD	S	K	1	2	3	4	5	6	7	8
1. Sex						-							
2. Age	42.19	(41.08, 43.33)	11.09	−0.12	−1.11	−0.31 **	-						
3. Org. Tenure	10.99	(10.08, 11.91)	9.04	0.82	−0.35	−0.19 **	0.67 **	-					
4. Avoidant WA	2.85	(2.71, 2.99)	1.39	0.69	−0.24	0.02	−0.12 *	−0.13 *	0.83				
5. Secure WA	4.15	(4.01, 4.28)	1.38	0.22	−0.73	−0.01	0.10	0.13 *	−0.55 **	0.87			
6. Preoccupied WA	2.30	(2.17, 2.44)	1.33	1.27	1.10	−0.10 *	−0.02	−0.04	0.65 **	−0.50 **	0.88		
7. Perceived comfort	3.29	(3.21, 3.36)	0.74	−0.33	−0.43	−0.07	0.08	0.02	−0.51 **	0.56 **	−0.41 **	0.78	
8. Perceived stress	2.72	(2.67, 2.78)	0.57	0.30	0.27	0.09	−0.22 **	−0.17 **	0.37 **	−0.19 **	0.38 **	−0.27 **	0.73

Note: *N* = 379; * = *p* < 0.01; ** = *p* < 0.001; SD = Standard deviation; S = Skewness; K = Kurtosis; LL = lower limit; UL = upper limit.

**Table 3 ijerph-20-05377-t003:** Mediating role of perceived comfort with secure workplace attachment as predictor.

	Perceived Comfort				Perceived Stress			
			Bootstrap 95%CI				Bootstrap 95%CI	
	*β*	SE	LL	UL	*β*	SE	LL	UL
Secure workplace attachment	0.55	0.04	0.47	0.64	−0.04	0.06	−0.15	0.07
Perceived comfort					−0.22	0.06	−0.34	−0.11
*Covariates*								
Sex	−0.06	0.04	−0.15	0.03	0.01	0.05	−0.09	0.11
Age	0.01	0.04	−0.08	0.10	−0.18	0.05	−0.29	−0.09
	R² = 32%				R² = 11%			
	F(3, 375) = 58.49, *p* < 0.001				F(4, 374) = 11.98, *p* < 0.001			

**Table 4 ijerph-20-05377-t004:** Mediating role of perceived comfort with avoidant workplace attachment as predictor.

	Perceived Comfort				Perceived Stress			
			Bootstrap 95%CI				Bootstrap 95%CI	
	*β*	SE	LL	UL	*β*	SE	LL	UL
Avoidant workplace attachment	−0.50	0.04	−0.58	−0.41	0.28	0.05	0.17	0.38
Perceived comfort					−0.11	0.05	−0.21	−0.01
*Covariates*								
Sex	−0.06	0.05	−0.14	0.04	0.02	0.05	−0.07	0.11
Age	0.01	0.05	−0.09	0.10	−0.16	0.05	−0.26	−0.07
	R² = 26%				R² = 17%			
	F(3, 375) = 44.41, *p* < 0.001				F(4, 374) = 19.68, *p* < 0.001			

**Table 5 ijerph-20-05377-t005:** Mediating role of perceived comfort with preoccupied workplace attachment as predictor.

	Perceived Comfort				Perceived Stress			
			Bootstrap 95%CI				Bootstrap 95%CI	
	*β*	SE	LL	UL	*β*	SE	LL	UL
Preoccupied workplace attachment	−0.41	0.05	−0.50	−0.32	0.32	0.05	0.22	0.41
Perceived comfort					−0.11	0.05	−0.21	−0.02
*Covariates*								
Sex	−0.09	0.05	−0.19	0.00	0.05	0.05	−0.04	0.14
Age	0.05	0.05	−0.05	0.14	−0.18	0.05	−0.27	−0.09
	R² = 18%				R² = 20%			
	F(3, 375) = 27.93, *p* < 0.001				F(4, 374) = 23.49, *p* < 0.001			

## Data Availability

The data presented in this study are available on request from the corresponding author.

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
