# Peer review of "The Mediating Role of Perceived Comfort between Workplace Attachment Style and Perceived Stress"

_ijerph, 2023, doi:10.3390/ijerph20075377_

Round 1

Reviewer 1 Report

The authors investigated the topic of work-related stress from the perspective of attachment theory. They discussed the relationship between the classical attachment theory and the place-related attachment with an emphasis on the workplace. Authors identified the lack in current literature, i.e., they stated no prior studies investigated the attachment style and stress relationship. They provided sufficient support for their hypotheses, both theoretical and empirical. The paper is valuable and interesting; the study was conducted properly (as for the observational plan), and the analyses were performed correctly. The main issue is the study design, which includes only a single measurement (at one point in time) and convenience sampling, but the Authors discussed this honestly.

The following issues must be addressed:

* The authors formed three series of hypotheses. However, no hypotheses addressed the mediation itself (only parts of this relation), while the mediation-type of relation is listed in the title and described in the paper. I suggest adding H4 to integrate this. 

* No details are provided regarding the sampling plan (only an overall type is listed). The authors should describe the recruitment process and the sources of sampling to allow others to analyze the potential bias or to replicate the procedure.

* Please calculate and report CIs and further variability for the main measured variables' summary statistics (given the convenience sampling, this should be required).

* No statistical power for the given sample size and obtained effect sizes is reported. If you did not plan sample size before the study, please report sample size analysis in the post hoc approach (obtained power).

Author Response

The authors formed three series of hypotheses. However, no hypotheses addressed the mediation itself (only parts of this relation), while the mediation-type of relation is listed in the title and described in the paper. I suggest adding H4 to integrate this. 

Thank you for this suggestion. We added the following mediation hypotheses.

H4a: Perceived comfort mediates the relationship between secure workplace attachment and perceived stress

H4b: Perceived comfort mediates the relationship between avoidant workplace attachment and perceived stress

H4c: Perceived comfort mediates the relationship between preoccupied workplace attachment and perceived stress

No details are provided regarding the sampling plan (only an overall type is listed). The authors should describe the recruitment process and the sources of sampling to allow others to analyze the potential bias or to replicate the procedure.

We modified the "method" paragraph and reported the following:

Employees was recruited online through posts on various professional (e.g., LinkedIn) and nonprofessional (e.g., Facebook) social media platforms. The only inclusion criterion for answering the questionnaire was that employees had been working for at least one year in the same organization.

Please calculate and report CIs and further variability for the main measured variables' summary statistics (given the convenience sampling, this should be required).

In table 2 we added 95% confidence intervals for mean

No statistical power for the given sample size and obtained effect sizes is reported. If you did not plan sample size before the study, please report sample size analysis in the post hoc approach (obtained power).

Thank you for this suggestion. We added:

Post hoc power analysis was performed using the statistical test named “Linear multiple regression: Fixed model R2deviation from zero”. Six predictors, 379 participants, a medium effect size of 0.15, and an α level = .05 were set. The analysis reported a power of .99.

Reviewer 2 Report

In the introduction, the theoretical and practical aim and importance of the research should be emphasized more broadly. The practical implications of the research should be further developed in the conclusion.

Author Response

In the introduction, the theoretical and practical aim and importance of the research should be emphasized more broadly.

Thank you for this suggestion. We modified introduction section end we added:

In light of these elements, this paper aims to investigate, for the first time, the relationship between workplace attachment styles and perceived stress, as well as the potential mediating role of employees’ assessment of the quality of their work environment – which we operationalized as their satisfaction with perceived comfort. From a theoretical point of view, this work, therefore, fits into that body of work in environmental psychology that recognizes the workplace (and the intentionality of it) as a resource of employees to cope with states of stress.   

The practical implications of the research should be further developed in the conclusion.

Thank you for this suggestion. We modified conclusion section and we added:

Specifically, we suggest that management should provide employees with a place that, on the one hand, meets employees' demands for comfort and, on the other hand, is capable of meeting their needs, this would allow them to develop positive self-representations that contribute to the structuring of a workplace secure attachment.

Reviewer 3 Report

This paper presents how previous research has shown how important attachment theory is for understanding workplace stress and the effects of employees' perceptions of their physical and spatial work environments on their health. The places of overlap between Bowlby's attachment theory and the place attachment theory used by environmental psychologists are the subject of several works, including this one. The authors hypothesized that a secure attachment style at work would be connected negatively with stress perception. (and vice-versa for insecure workplace attachment styles). Theoretically, perceived comfort could mediate these outcomes. The PROCESS macro was used to test hypotheses.
Statistical processes were used to test the hypotheses.
The manuscript presents the information clearly and extensive.

Author Response

Thank you!